# Free-Stall Use and Preferences in Dairy Cows: A Case Study on Neck Rails Covered by Foam

**DOI:** 10.3390/ani9100772

**Published:** 2019-10-09

**Authors:** Marek Gaworski

**Affiliations:** Department of Production Engineering, Institute of Mechanical Engineering, Warsaw University of Life Sciences—SGGW, Nowoursynowska str. 164, 02-787 Warsaw, Poland; marek_gaworski@sggw.pl; Tel.: +48-22-593-45-83

**Keywords:** cow, free-stall system, foam, neck rail, preference, row

## Abstract

**Simple Summary:**

Neck rails are used in many free-stall barns; they are intended to keep cows standing at the rear of the stall such that faeces and urine are more likely to fall in the alley rather than on the stall surface. However, cows can come into contact with the neck rail when entering the stall or when standing up. This study tested the effect of covering neck rails with a protective foam surface, by allocating stalls to control and foam conditions using a crossover design. There was no effect of the foam treatment on the time that stalls were occupied for lying. Considering the time of individual stall occupation and the distribution of the stalls in the pen, it was found that cows preferred one of the two lying stall rows. Analysis of variance showed a significant difference between the time of stall occupation for two different rows in the pen, including the time for lying as well as the time of standing with two and four hooves in the stall. Results from this study can be used to support the design process for free-stall barns.

**Abstract:**

This study tested the effect of neck rails equipped with and without foam on stall usage and preference by dairy cows. The hypothesis of the experiment, that cows prefer lying stalls with foam in comparison to stalls without foam, was rejected. There was no significant difference (*p* > 0.05) in lying time and time spent standing with two and four hooves in the lying stalls between the two treatment groups. Considering the time of individual stall occupation and the distribution of stalls in the pen, cows showed a preference for one of two lying stall rows. Analysis of variance showed a significant difference (*p* < 0.001) between the time of stall occupation for the two different rows in the pen, including the time spent lying and the time spent standing with two and four hooves in the stall. Lying time per stall in the preferred row (near the feeding alley) in comparison with the alternative row was 580 ± 101 min·d^−1^·stall^−1^ versus 50 ± 28 min·d^−1^·stall^−1^, respectively. These results can support a design process for new barns with a free-stall housing system for dairy cattle.

## 1. Introduction

A free-stall housing system is one solution implemented to improve cow comfort. To better recognize and assess cow comfort conditions in free-stall barns, many investigations have been conducted, including studies assessing technical and technological aspects of free-stall design and different barn management practices. In the field of technical infrastructure, focused on housing design, Tucker et al. [1] presented the effects of free-stall dimensions on the preference and stall usage in dairy cows, illustrating that providing cows with wider free stalls increased lying times, likely because cows had less contact with the partitions in larger stalls. On the other hand, narrow stall partitions can reduce the risk of cows soiling the stall. Tucker et al. [2] found that brisket boards can reduce free-stall use, suggesting that cows preferred to have their body fully supported by the bedded area. Ruud and Bøe [3] showed the effects of flexible and fixed partitions in free stalls on lying behaviour and cow preference. Drissler et al. [4] measured the changes in lying behaviour when groups of cows had access to free stalls with different amounts of sand bedding and found that lying times reduced with decreasing bedding amounts. Gaworski et al. [5] showed the effects of stall design on dairy cow behaviour. Ceballos et al. [6] indicated advantages arising from improved stall design using 3-D kinematics to measure space use by dairy cows when lying down. 

One important element in free-stall design is the position of the neck rail. The neck-rail position in a free stall affects standing behaviour and udder and stall cleanliness [7]. Moreover, neck-rail placement has been considered in regard to free-stall preference, use, and cleanliness [8]. This illustrates the stall-design paradox, that is, a restrictive neck-rail placement increases lameness but improves udder and stall hygiene [9]. 

There is a high probability that dairy cows come into contact with different parts of a free stall (e.g., contact with stall bedding). The bedding quality [10,11], the type of bedding material [12], the stall lying surface [13], and dairy cattle motivation to lie down [14] can all affect lying time. Dairy cows can also come into contact with other parts of a free stall, most importantly the neck rail. The location of the neck rail can play an important role in stall usage, for example, lower neck rails reduce stall usage by cows [15]. However, neck rails can have both negative and positive effects [8]. When the neck rail is in the less restrictive position (i.e., further away from the rear curb), cows stand with all four hooves placed within the stall, and the top of their necks can gently touch the neck rail. However, in a more restrictive neck-rail position, contact with the neck rail can become a source of discomfort for an animal because in many cases the animal hits the neck rail. If the bottom side of the neck rail has a worn appearance, cows are likely hitting their neck against the neck rail when rising from the stall surface. In pasture, cows rise and end up standing 60 to 90 cm in front of where they were lying. Therefore, wherever the neck rail is placed, it will be in the way of the cow, even if it is “floating” or is made of some material other than metal [16]. 

While there is a substantial amount of work investigating neck-rail placements, more work studying neck-rail material and the effects on cow behaviour and lying stall preferences is required. The objective of this study was to assess the effect of neck-rail position on cow behaviour, specifically free-stall usage and cow preferences. This study investigated different positions of the neck-rail bottom (as a result of installed additional elements, namely foam on the neck rail) above the bedded surface and its effect on cow behaviour and preferences. It was anticipated that there would be differences in stall use based on the considered options of the neck-rail bottom positioning, as direct contact between the cow and a neck rail with foam would be more cushioning. Additionally, preferences for specific lying areas within the free-stall pen were investigated. Studies assessing animal preferences can be used to improve housing conditions and cow comfort in dairy production. 

## 2. Materials and Methods

This experiment was conducted at the University of British Columbia’s Dairy Education and Research Centre—Agassiz (Agassiz, BC, Canada), in December when the daily average outside air temperature varied between 2 and 7 °C. The animals were cared for according to the guidelines of the Canadian Council of Animal Care [17]. 

Six Holstein dry (i.e., non-lactating) cows were assigned to the study. Cows were selected from a pool of dry cows based on no signs of clinical lameness and those with furthest expected calving dates. There was a difference of 11 days between cows with the earliest and latest calving dates. Dry cows were selected for this experiment in accordance with the number of available cows on site. Cows averaged (mean ± SD) 2.83 ± 1.17 lactations, with a body weight of 635 ± 70 kg and height of 145 ± 6 cm. No animals showed signs of illness during the study.

### 2.1. Housing and Management

Cows were housed in one pen in a naturally ventilated wood-frame, free-stall barn with curtained sidewalls. The test pen contained twelve free stalls, configured in two rows facing in opposite directions (Figure 1). The pen was equipped with stalls divided by Y2K-style (Artex, Langley, BC, Canada) partitions and measured 1.28 m wide centre to centre. The outermost stalls from one side in each row were restricted by a pen fence, whereas the outermost stalls from the second side in each row were restricted only by partitions that separated the stalls and a passing area connecting the scraper alleys.

Stalls were bedded with washed river sand and raked to a uniform level in line with the rear curb and the brisket board at the beginning of the study. During the period of the experiment, sand on the lying stalls was not filled in. This approach was in accordance with the general rule applied on the farm where new sand was added to the stalls each 14th to 18th day. The experiment was run for 18 days, so it was justifiable not to add new sand, all the more so because the cows before the experiment were housed in the barn with sand added every two and a half weeks. Twice daily (during morning and evening feedings), faeces were removed from stalls using rakes. The alleys were covered with a textured rubber surface and were scraped automatically 6 times/d. Crossover alleys were grooved concrete and scraped manually twice daily.

Dry (i.e., non-lactating) cows were fed ad libitum a total mixed ration (TMR) consisting of 30.54% corn silage, 46.52% alfalfa hay, 13.71% concentrate mix, and 9.23% rye grass seed straw. Fresh feed was provided twice daily at 08:00 am and 17:00 pm. Feed was pushed up at 11:00 am, 19:00 pm, and 22:30 pm. Water was available ad libitum from one self-filling water bin (Insentec, Marknesse, the Netherlands) located in the area of the feeding alley at the right corner of the pen (Figure 1). The water bin was cleaned once per week.

### 2.2. Experimental Design and Treatments

The 12 lying stalls in the test pen were alternately assigned as controls (i.e., 6 stalls with a typical steel neck rail) or to the foam condition (6 stalls with the neck rail covered by protective polyurethane foam—density of 25 kg·m^−3^, 8 cm thick, and 90 cm in length positioned in the middle of the neck rail). At the beginning of the experiment, the vertical distance between the bottom foam-covered rails and the bedded surface was 116 cm, whereas the distance between the bottom of the control rail and the bedded surface was 124 cm. Sand bedding was manually raked level at the time of morning and evening feedings when most cows were standing in the feeding alley.

Before the experiment, cows were housed together for one week with all neck rails in the control condition. The foam was then applied to the test stalls and cows were allowed two days to acclimatize, and then cows were video-recorded for one full week. At this point, treatment conditions were reversed (i.e., stalls originally assigned to the foam condition became controls and vice versa) and cows were allowed two days to acclimatize and then video-recorded for another full week. In each row, foam was placed on every second stall, with foam being added to the first stall in row I and starting in the second stall in row II (Figure 1). The order of treatment was reversed for the second week of the experiment.

The foam was connected to the neck rail by wrapped tape in the centre and at both ends of the foam to the neck rail. The use of tape decreased the tendency for the foam to change position during the experimental period. Regardless of such solutions, the proper position of foam on the neck rails was checked, and if necessary, corrected each day. In addition to the foam, the neck rails in the lying stalls in the experimental pen were characterized by some additional features. There was no continuous neck-rail pipe connecting all six stalls in one row, but rather three independent neck rails, that is, one neck-rail pipe per two lying stalls (see Figure 1). Each neck rail was connected by only one point to the top part of the partition. This meant that each stall was equipped with a neck rail installed at only one point. As a result, in addition to one fastened end, the neck rail in each stall had one free end.

The video camera (WV-BP330, Panasonic, Osaka, Japan) was positioned 8 m above the pen, and was attached to a digital video recording system (Genetec Inc., Saint-Laurent, QC, Canada). A red light bulb (100 W) was hung 8 m above the pen to facilitate video recording at night. Cows were marked with unique symbols using hair dye to identify individuals and the stalls were numbered.

### 2.3. Behaviour

The behaviour of the cows was video-recorded for 24 h/d, 7 d/week. Video recordings were scan sampled at 10 min intervals to quantify stall use and cow positions in the stalls, including lying, standing with two hooves in the stall, or standing with all four hooves in the stall. Data were used to calculate the total time per stall (the experimental unit) as well as per cow for each of these activities. Cows were never observed lying down outside of the stall. Free-stall design features help to maintain stall cleanliness, but such effects were not analyzed.

### 2.4. Statistical Analysis

The stall was the experimental unit so only stall-based results were statistically analyzed using Statistica v.13 software (StatSoft Polska, Cracow, Poland). Normality was evaluated by the Shapiro–Wilk test for normality. Cow-based results are provided for descriptive purposes only. The time that the stalls were occupied in total, and separately by cow position (lying, standing with two and four hooves in the stall), was analyzed using a mixed model, testing the effect of treatment and stall row, with stall identity included as a random effect. Significance was declared at α = 0.05.

## 3. Results

There was no effect of treatment on the amount of time a stall was occupied. Stalls with the foam were occupied by a lying cow for 13.2 min·d^−1^·stall^−1^ longer time than stalls in the control condition (no foam covering). Similarly, there was no effect of the foam treatment on time cows spent perching with their front two hooves in the stall; the difference between foam treatments was 1.4 min·d^−1^·stall^−1^. Standing with all four hooves in the stall included 0.2 min·d^−1^·stall^−1^ difference for stalls with and without the foam covering (Table 1).

Lying time in stalls with foam was approximately 4.3% longer than the time spent by cows lying in stalls without foam, but the difference was not significant. Time spent standing with two and four hooves in the lying area was approximately 1.5% longer for cows in stalls without foam in comparison to cows with foam, however the difference was not significant.

The data concerning time of stall occupation (Table 1) did not show significant differences between stalls with and without foam placement on the bottom part of the neck rail. However, other aspects of stall usage were observed in the current study. Including the 12 stalls in the experimental pen, the time of individual stall occupation was calculated for the two-week period, expressed in (min⋅d^−1^⋅stall^−1^) and presented in Figure 2, Figure 3 and Figure 4 for three considered cow positions, lying (Ly), standing with two hooves in the stall (S2) and standing with four hooves in the stall (S4).

The distribution of the data given in Figure 2, Figure 3 and Figure 4 shows that stalls with the longest time of occupation by cows are concentrated in the area with stall Nos. 7 to 12. Stalls 7–12 create one full row (Figure 1) and are alternatives for the second row with stall Nos. 1 to 6, which is located in the same pen but on the opposite side. The same pattern was observed for lying, standing with two hooves in the stall and standing with four hooves in the stall. Analysis of variance showed a significant difference (*p <* 0.001) between the time of stall occupation for two rows in the pen and each considered activity (lying, standing with two and four hooves in the stall). A graphical presentation of the difference between the two rows (Figure 5) shows that the time of stall occupation for lying (Ly) in one row (with stalls 7 to 12, that is, row II; see Figure 1) was nearly 12-fold longer than in the alternative row (with stalls 1 to 6, that is, row I; see Figure 1).

## 4. Discussion

The time that stalls were occupied for lying and standing did not vary with the foam treatment. However, the foam covering reduced the space between the neck rail and the bedded surface. Previous work has shown that neck rails placed closer to the bedded surface of the stall [8] and closer to the rear curb reduced the time cows spend standing fully in the stall and increased perching in the stall [7]. It is possible that the lower neck-rail position diminished any benefit of the foam; future work should include treatments that control for the height of the neck rail.

The results given by a descriptive study [15] indicated that lower neck rails could reduce stall usage. However, Tucker et al. [8] demonstrated that neck-rail placement affects standing behaviour in the free stall. For stalls with the lower position of the neck rail, the cows spent less time standing with all four hooves in the stall. In this study, including the additional observation of cows with front hooves in the stall, the longest time of standing behaviour was found in the stalls with no neck rail. The time spent by cows lying in the stall did not differ significantly among the four investigated neck-rail positions (neck-rail height). According to research carried out by Fregonesi et al. [7], when the neck rail was positioned further from the rear curb, cows spent less time standing with their front two hooves in the lying area and more time standing with all four hooves in the lying stall. Lying and standing behaviours are often used both as a sign of well-being in dairy cattle and to evaluate the quality of stalls [18].

There are two key parameters in the assessment of interactions between cows and neck rails, namely neck-rail height (above the lying surface) and distance to the inner side of the rear curb. Tucker et al. [8] included the following neck-rail heights in their experiment—102 cm (40 in), 114 cm (45 in), 127 cm (50 in), and a “no neck rail” option. On the other hand, Fregonesi et al. [7] tested one constant height of 125 cm above the bedded surface at five neck-rail positions, being 130, 145, 160, 175, and 190 cm from the rear curb. In the current experiment, the investigated heights of the neck rail above the bedded surface were 116 cm and 124 cm, whereas the distance between the neck rail and the inner side of the rear curb was 200 ± 4 cm (mean ± SD for 12 lying stalls in the investigated pen).

It is necessary to indicate that the height of the neck rail was measured at the beginning of the experiment, but the level of the sand as a bedding material—through the effects of cow activities and natural usage—could have decreased during the following days [4]. A decrease in the sand level changed the lying conditions within the experimental period. The reaction of cows to lower amounts of sand in free lying stalls may be expressed by a shorter lying time [4]. This was confirmed by the results of the current experiment; stalls with neck rails covered by foam were occupied for a shorter time by lying cows in the second week in comparison with the first week. The time varied by 10.8%, but this difference was not significant. Only for the time of standing with four hooves in the lying stalls with a neck rail covered by foam was the difference for first and second weeks significant, with a longer time, in min·d^−1^·stall^−1^, in the second week than in the first one. Such results show that decreasing the time of stall occupation for lying within the observation period can be associated with longer times of standing activities in the stalls.

The experiment originally aimed to find differences between stalls with two options for the neck-rail surface. However, it was difficult to compare how long stalls were occupied in the current experiment, as done by [5]. The experimental pen consisted of 12 stalls and only 6 cows, which was compatible with the general rule for preference tests. On the other hand, a higher number of stalls than cows in one pen reduced the possibility for comparing stall loading between experiments.

Including data for the observation period of 14 days, the mean lying time for the group of six cows was (mean ± SD) 629.6 ± 182.1 min·d^−1^·cow^−1^. Considering the results for individual cows, the longest lying time was (mean ± SD for 14 days) 801.4 ± 61.8 min·d^−1^·cow^−1^, whereas the shortest one amounted to (mean ± SD for 14 days) 359.3 ± 151.8 min·d^−1^·cow^−1^. In the group of six cows, the longest and shortest observed times of standing with two hooves in the stall were (mean ± SD for 14 days) 597.1 ± 105.3 min·d^−1^·cow^−1^ and 44.3 ± 18.7 min·d^−1^·cow^−1^, respectively. For standing with four hooves in the stall, the longest and shortest times concerning individual cows amounted to (mean ± SD for 14 days) 93.6 ± 36.5 min·d^−1^·cow^−1^ and 7.9 ± 8.0 min·d^−1^·cow^−1^, respectively.

The cross-sectional studies on dairy farms in the northern US indicated that average lying times among high-producing cows were 10 to 11 or more hours per day [19]. The average lying time of cows included in the experiment was approximately 10.5 h·d^−1^ (629.6 min·d^−1^·cow^−1^), which coincided with the cited research. Other studies have indicated more variation in lying behaviour among individual cows within farms rather than across farms [20]. The possible variation in lying behaviour among individual cows within a group was confirmed by the current study. The difference between cows with the shortest and longest lying times was more than double.

Additional observations on cow lying behaviour were obtained. Cows were lying for approximately 7% longer in the first stage (days 1–7) than in the second (days 8–14). The opposite situation occurred for the two types of standing behaviour—in the second stage (days 8–14), the time of standing was longer than in the first stage (days 1–7) by approximately 11.5% for standing with two hooves and 18% for standing with four hooves in the stall.

While cows did not show preferences for stalls with a foam covering on the neck rail, the cows did show a preference for one row of stalls located in the experimental pen. Such results confirm that it is possible to find further research areas where cows show behavioural activities, need, and response to their environmental conditions. Stalls at the feeding alley from the pen were more likely to be occupied for all three behaviours monitored than the stalls at the back of the pen. Thus, the question arises: why did cows prefer one of two rows in the pen? This preference may have been driven by cows preferring to face in the direction of the feeding alley, perhaps so they could monitor activity in the feeding area, including times when new feed was dispensed or old feed pushed closer to the cows. Between the preferred rows with lying stalls and the feeding alley was only the scraper alley (Figure 1), so there were no barriers to good visibility of the feeding alley. In addition, these stalls were located more closely to the feeding alley. The row with preferred stalls was located between the two scraper alleys, whereas the other row included stalls adjacent to the fence near the walking alley along the wall with curtains. The walking alley was used to transfer cows from other pens in the barn to the milking parlour two times per day (in the morning and afternoon). Schmisseur et al. [21] reported that a proportion of cows show preferences for specific stalls. Friend and Polan [22] demonstrated that dominant cows occupied stalls near the entrance in one row and near the middle in the other; moreover, the location of stalls most frequently used by each cow in 2 rows of 10 could be described by a quadratic function of cow social rank. Wagner-Storch et al. [23] found that cow lying and stall occupancy percentages were highest for stalls located not at the end of a section and on the outside row and varied by the base type for the time the cows were exposed to the stall or inside barn temperature. Gaworski et al. [5] found that cows preferred stalls closest to the feeding alley. In combination, these results indicate that stall location needs to be controlled for in any study examining factors affecting stall use.

Knowledge resulting from cow observations becomes part of future studies aiming to improve new barn designs suitable for improved animal welfare; these studies are needed to determine daily cow behavioural activities [24].

The period of observation in the experiment included only two weeks, at the end of autumn (in December). On the other hand, there could be a possibility of seasonal variation in the area preferences of dairy cows in free-stall housing [25,26] and the eligibility of lying boxes at different Temperature Humidity Index (THI) levels in a free-stall barn [27]. Thus, it seems to be of value to develop further observations of cow preference during other seasons.

The conducted experiment and obtained results provide insight into stocking densities and stall usage. According to Fregonesi et al. [28], overstocking reduces lying time in dairy cows; when animals were overstocked, they spent less time lying down, whereas the time standing outside the free stalls was longer. However, Wang et al. [29] found that overstocking did not cause a negative effect on behaviour, productivity, or comfort indices of cows in that study, whereas understocking contributed to the natural behaviours of cows including lying, feeding, and rumination. It can be considered that the results of such natural behaviour in the presented experiment with understocking (6 cows versus 12 stalls) was a preference expressed by the cows for stalls in one row. There is a possible indication of other benefits coming from the observation of a cow herd kept in an understocking regime. Talebi et al. [30] found that reduced stocking density mitigates the negative effects of regrouping in dairy cattle.

Further research can lead to a better understanding of how stall location affects cow preference for different stall bases [23]. An improved understanding of stall popularity may prove to be a fruitful direction for improving return for the large investments made in housing facilities [28]. The preferences of dairy cows for stalls and their locations can be significant information for use in barn design to improve dairy production on the farm.

## 5. Conclusions

Adding foam to the bottom of the neck rail had no effect on stall use and preferences, suggesting that neck-rail softness has little effect on stall use, or that other types of design need to be tested. Stall use was found to be higher for stalls located near the feed alley as compared to further away from the feed alley, showing the effect of stall location on stall occupancy.

## Figures and Tables

**Figure 1 animals-09-00772-f001:**
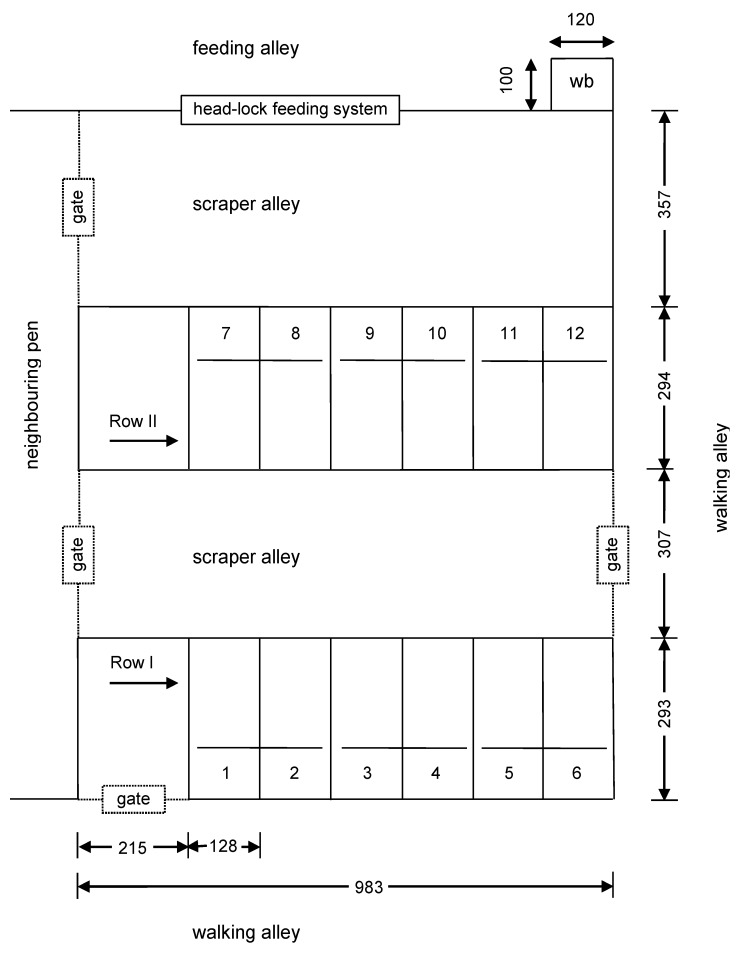
Configuration of the pen used for the experiment with modified neck rail taking into account row numbers (I and II), stall numbers (1–12), outside details, and dimensions (cm); wb, water bin.

**Figure 2 animals-09-00772-f002:**
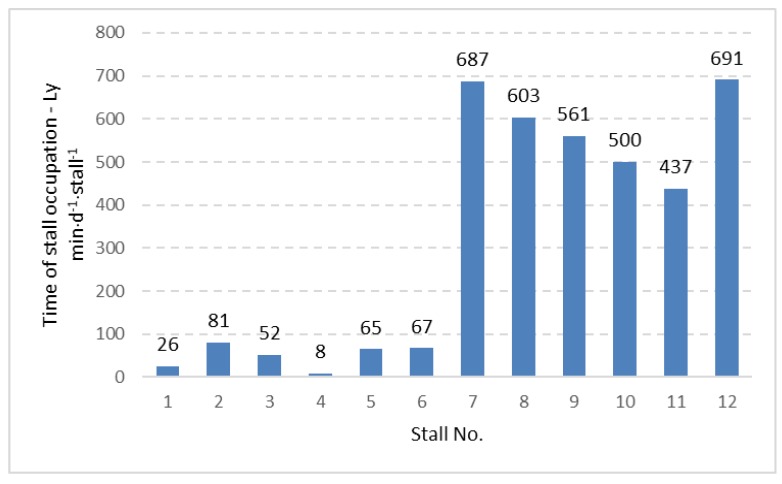
Distribution of time of occupation for lying (Ly) for individually considered stalls in the investigated pen.

**Figure 3 animals-09-00772-f003:**
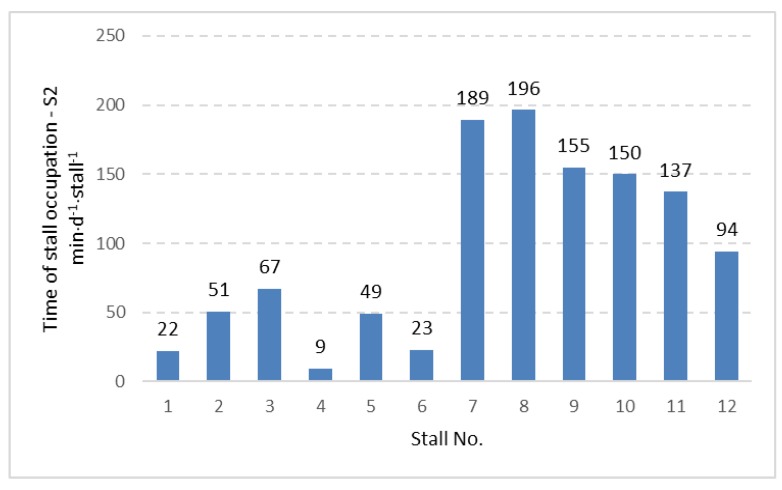
Distribution of time of occupation for standing with two hooves in the stall (S2) for individually considered stalls in the investigated pen.

**Figure 4 animals-09-00772-f004:**
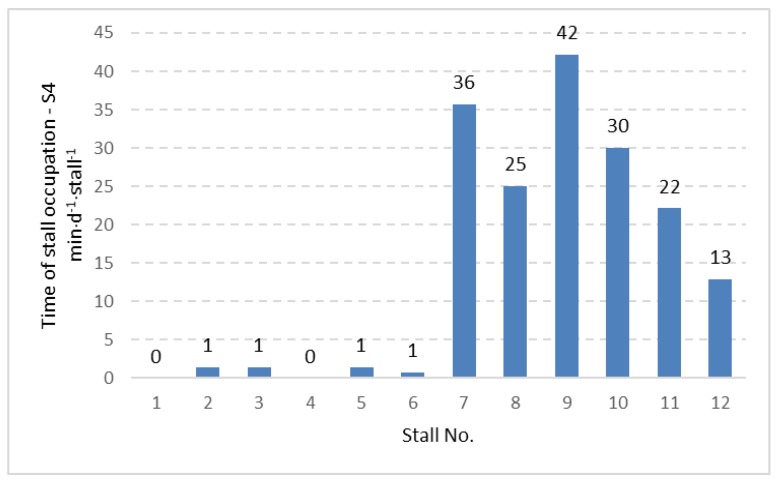
Distribution of time of occupation for standing with four hooves in the stall (S4) for individually considered stalls in the investigated pen.

**Figure 5 animals-09-00772-f005:**
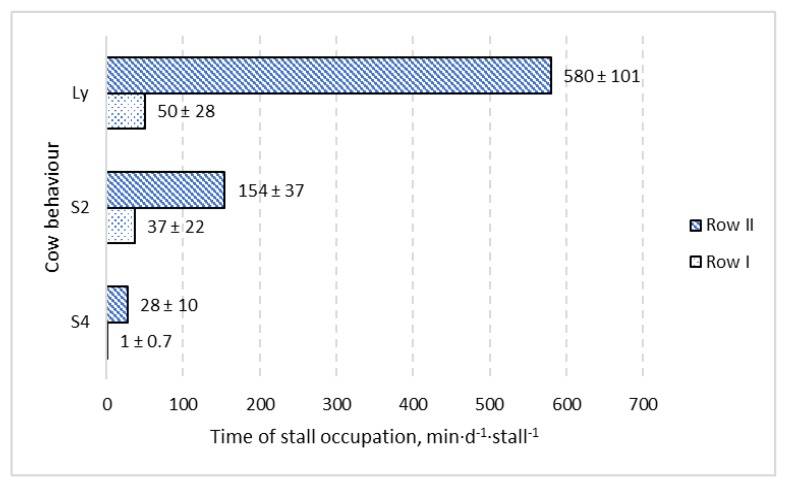
Comparison of time of stall occupation (mean ± SD) for two alternative rows (row I and row II) including three kinds of cow behaviour: lying (Ly), standing with two hooves in the stall (S2) and standing with four hooves in the stall (S4).

**Table 1 animals-09-00772-t001:** Time of stall occupation including three kinds of cow activities in the experiment with neck rail (nr), in (min·d^−1^·stall^−1^).

Behaviour	Treatment	Mean	± SD	Minimum	Maximum
Lying	nr with foam	321.4	51.5	188.3	403.3
	nr without foam	308.2	46.2	220.0	395.0
Standing with 2 hooves in the stall	nr with foam	94.6	23.6	40.0	136.7
	nr without foam	96.0	28.1	50.0	170.0
Standing with 4 hooves in the stall	nr with foam	14.3	6.1	3.3	21.7
	nr without foam	14.5	6.3	1.7	23.3

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
