# Peer review of "Free-Stall Use and Preferences in Dairy Cows: A Case Study on Neck Rails Covered by Foam"

_animals, 2019, doi:10.3390/ani9100772_

Round 1
Reviewer 1 Report
Line 63- "appropriate" is not a fear term for neck-rail position. The neck-rail position represent a paradox for producers. I would suggest "less restrict position"
Line 212 - I think height of neck-rail was a very important variable in this behaviour and preference experiment. The author could explain further why it was not under control, please?
Line 249 - Do you think that an experimental design with two restrict and one choice phases would give clearer behaviour and preference results? Please, explain it.
Author Response
Dear Reviewer,
Thank you for the prepared review of the manuscript. My answers for the indicated problems in the manuscript are as following:
Regarding „Line 63”: change of the words was done
Regarding „Line 212”: I am fully agree with reviewer that height of neck-rail was a very important variable in the behaviour and preference experiment. Lying area with sand as a bedding material is characterized by dynamic changes in sand level as a result of individual activities of cows visiting lying stalls. These changes can be observed after each cow visit. Including some differences in cow height (wither height), freedom of lying stall choice and the changes in sand level during 24 hours, the relationship neck-rail height – individual cow is the effect of many variables. In my opinion it was possible to measure height of neck-rail each 24 hours but for such measurements it is necessary to choose the time, when all stalls are not occupied. For experimental group of dry cows it was difficult. When I looked through again the collected data I haven’t found the same time each day, when all lying stalls weren’t occupied. It was intended to not disturb cows by additional direct measurements during their stay in the experimental pen. In my opinion it is possible to design simply device to more easy and faster way (than use of measuring tape) find distance between neck-rail and sand surface at lying area. Such device can base on wheel (run along the neck-rail bottom) with mechanism starting measuring laser signal each full turn of the wheel. The device can be equipped with panel to record all data concerning distances between neck-rail and sand surface along the stall width. This device can be used for fast measurements in each lying stall with less disturbances in the pen with animals.
Regarding „Line 249”: Thank you for the disputable question. In my opinion, an experimental design with two restrict and one choice phases can give clearer behaviour and preference results for the considered case study. I think that the results can be clearer, when disturbing factors are eliminated. There are many factors, which in positive and negative way decide about behaviour and preference results, especially in the experiments with dairy cows. So in my opinion animal choice between more options found in production / barn conditions can show wider spectrum of animal behaviour and preferences. There are recognized negative factors restricting lying behaviour of cows, like high moisture of bedding material, low level of bedding material in the lying stall, too short diagonal distance between neck-rail and rear curb and the others. Developing the discussion about the behaviour and preference results there is possible to put the question: which negative factors influencing cow behaviour and preference can be more and less negative ? The same question is possible to put for the positive factors.
Reviewer 2 Report
Dear Editor,
the paper is well written and of interest for the readers of the journal.
The study investigated the effects of modifying position and materials of neck-rails on cow behaviour. The paper should be improved before possible publication in the journal.
I suggest to the Author a little change to the title because it is too generic, especially for the word 'modification', and maybe is also a little confused. The Authors could use a synthetic general title and then a subtitle more specific to the case analysed (i.e., the use of foam).
Line 86: the Author should explain to the readers why he/she considered dry-dairy cows and far from calving.
English should be a little improved because some sentences are not well structured or too long (i.e., lines 57-62; lines 67-70; lines 83-84; lines 95-98...and so on).
The Author should avoid personal form such as 'I' (line 87).
Please use standards for measurements units (line 84: degree-> °C).
Use as label Figure I instead of Scheme I (line 114) and add this change to the whole document when referring to Scheme I.
Author Response
Dear Reviewer,
Thank you for the prepared review of the manuscript. My answers for the indicated problems in the manuscript are as following:
Title of the paper was changed into “Free-stall use and preferences in dairy cows: a case study on neck-rails covered by foam”.
Sentence explaining use of dry-dairy cows was given.
Some suggested sentences in the paper were improved.
“I” was removed from the paper, so there were prepared new sentences.
“°C” was included instead of “degrees Celsius”.
“Figure 1” was included instead of “Scheme 1”.